# Extruded-Calendered Sheets of Fully Recycled PP/Opaque PET Blends: Mechanical and Fracture Behaviour

**DOI:** 10.3390/polym13142360

**Published:** 2021-07-19

**Authors:** David Loaeza, Jonathan Cailloux, Orlando Santana Pérez, Miguel Sánchez-Soto, Maria Lluïsa Maspoch

**Affiliations:** Centre Català del Plàstic, Universitat Politècnica de Catalunya Barcelona Tech (EEBE-UPC), Av. d’Eduard Maristany, 16, 08019 Barcelona, Spain; alfonso.david.loaeza@upc.edu (D.L.); jonathan.cailloux@upc.edu (J.C.); m.sanchez-soto@upc.edu (M.S.-S.); maria.lluisa.maspoch@upc.edu (M.L.M.)

**Keywords:** post-consumer opaque PET, titanium dioxide, recycling, post-consumer PP, essential work of fracture

## Abstract

This work presents the experimental results of the mechanical and fracture behaviour of three polymeric blends prepared from two recycled plastics, namely polypropylene and opaque poly (ethylene terephthalate), where the second one acted as a reinforcement phase. The raw materials were two commercial degrees of recycled post-consumer waste, i.e., rPP and rPET-O. Sheets were manufactured by a semi-industrial extrusion-calendering process. The mechanical and fracture behaviours of manufactured sheets were analyzed via tensile tests and the essential work of fracture approach. SEM micrographics of cryofractured sheets revelated the development of in situ rPP/rPET-O microfibrillar composites when 30 wt.% of rPET-O was added. It was observed that the yield stress was not affected with the addition of rPET-O. However, the microfibrillar structure increased the Young’s modulus by more than a third compared with rPP, fulfilling the longitudinal value predicted by the additive rule of mixtures. Regarding the EWF analysis, the resistance to crack initiation was highly influenced by the resistance to its propagation owing to morphology-related instabilities during tearing. To analyze the initiation stage, a partition energy method was successfully applied by splitting the total work of fracture into two specific energetic contributions, namely initiation and propagation. The results revelated that the specific essential initiation-related work of fracture was mainly affected by rPET-O phase. Remarkably, its value was significantly improved by a factor of three with the microfibrillar structure of rPET-O phase. The results allowed the exploration of the potential ability of manufacturing in situ MFCs without a “precursor” morphology, providing an economical way to promote the recycling rate of PET-O, as this material is being discarded from current recycling processes.

## 1. Introduction

In the last decade, the excellent strength and durability of PET packaging drove the growth of the colored PET market. The base of some packages produced from colored rPET gives structural strength, which helps to protect goods from external agents. This led to the recent development and commercialization of white-opaque PET bottles (PET-O) used mainly for packing milk. This is a PET that has been made white and opaque by the addition of titanium dioxide (TiO_2_) between 5% and 15% wt.% [1]. As the fiber production in the textile sector is the principal end-use market of rPET [2], mechanical recycling issues have been rapidly reported when the recycled PET-O (rPET-O) are physically mixed with transparent rPET, because this addition leads to processability problems during fiber manufacturing [3]. That is, the presence of TiO_2_ retards PET strain hardening and stress-induced crystallization in the rubbery temperature range, which makes fiber manufacturing difficult [4]. On this basis, the development of new market opportunities for recycle PET-O (rPET-O) is receiving increasingly significant attention in order to target the European Union circular economy policy and its related plastic strategy by 2030 [5].

In view of the potential immediate future problem in waste management policies for this material, a European project (RevalPET) has been proposed that aims to design new formulations of plastic material with improved properties using rPET-O for applications not limited to the manufacture of fibers [6]. Among the proposals in the RevalPET project, the use of rPET-O as reinforcer in microfibrillar form of a recycled polypropylene (rPP) matrix has been explored. Because of the immiscibility between these polymers and their different melting temperatures, *T_m_*, the selection of processing conditions was considered in order to favor an in situ microfibrillation process, induced during the preparation of the blend and subsequent shaping of piece in order to generate microfibril reinforced composites (MFCs) [7,8].

Briefly, the technique to produce these MFCs is based on preparing a homogeneous polymer blend of the matrix A and the dispersed phase B. The *T_m_* of B should be at least 40 °C higher than the one of A, hence the processing temperature for blend preparation is selected as follows: *T*_melt blending_ > *T_m_*A and *T_m_*B. Prior to pelletization, the blend is drawn either in melted state (hot drawing) or already solidified (cold drawing), which transforms the spherical domains of B into highly oriented microfibrils with an increased slenderness. The pellets, with this “precursor morphology” are then used for the manufacture of the final parts, fulfilling the following thermal processing condition, in order to preserve and even increase the aspect ratio of the B-microfibrils: *T_m_*A < *T*_process_ < *T_m_*B. This type of approach to MFCs’ manufacturing has shown significant reinforcement effect in polyolefins blended with engineering virgin and recycled polymers such as PET, polycarbonate (PC), polyamides (PA), or polyolefins—mainly PE and PP [9,10,11]. Types of PP/PET blends have been widely analyzed thanks to its potential and economical way to produces MFCs [9,12,13].

It is well known that, in immiscible polymer blends, such as the one proposed here, the properties are determined by the morphology obtained, in particular, of the final characteristics of the domains of the minority component: average size, shape, distribution, and distance between domains [14]. Some authors have found that, in 70/30 wt.% PP/PET blends, a more homogeneous phase morphology is achieved by increasing the screw speed, but further increase caused agglomeration of PET droplets. On the other hand, the increase of PET content up to 40–50 wt.% reduced the storage modulus of blends [15,16]. Mechanically, it has been reported that the tensile and flexural strength linearly increased by the addition of rPET up to 40 wt.%. However, addition of rPET and the increased rPET content significantly decreased the impact strength [17,18] as well as the strain properties [19]. In recent years, the use of reinforcing elements in recycled blends has increased. rPET ranging from 10 to 20 wt.% has been used to analyze the effect on the incorporation of reinforcing fillers [20,21], increasing the mechanical properties in the blends. Tramis et al. showed an significant increase by up to an eightfold in the fatigue life of rPP by the incorporation of 20 wt.% of rPET-O [22].

For this reason, the study of the mechanical and fracture performance in these polymeric systems requires special attention because of the microstructure-related anisotropy and molecular orientation induced during the conformation step, such as the extrusion-calendering process. In this case, this anisotropic behaviour will be dictated not only by the conditions in the extrusion itself, but also by the subsequent calendering stage (calibrating, cooling, and collecting). When working with immiscible polymer blends in extrusion-calendering, the situation becomes even more complicated because there is a type of additional anisotropy produced by the orientation of the polymeric phases. The juxtaposition of the elongation and shear stress fields generated in the region located between the exit of the extrusion head and the contact with the first calendering roll.

As the proposed polymer blend in this study exhibited a post-yielding crack propagation, a straightforward method to analyze and quantify the fracture is the well-stablished essential work of fracture (EWF). This analysis technique allows evaluating mechanical work involved in both deformation events: the crack propagation plane (generation of new free surface) and in the region surrounding said plane during the tearing process, directly related to resistance to stress crack propagation. Briefly, it proposes a split of the specific total work of fracture (work by unit surface of the crack, *w**_f_*) in the specific essential work of fracture (*w**_e_*), which ascribes to aforementioned in-crack plane work, and in the specific non-essential work of fracture (*βw_p_*), which is related to the work by unit volume of material involved in the out-of-crack plane [23]. The interested reader can find more information regarding the EWF concept elsewhere [24,25].

In a previous work [26], a detailed comparison of the mechanical and fracture behaviour between unfilled rPET (or transparent rPET-T) and rPET-O was reported. On this basis, in the present study, the effect of rPET-O in three different concentrations on the mechanical and fracture behaviour of a post-consumer PP grade is discussed. In order to analyze the rPET-O response as a reinforcing element, its content was varied in a range less than 40%. The fracture behaviour of rPP/rPET-O blends was assessed following the EWF concept on calendered sheets manufactured using a semi-industrial extrusion-calendering process. The energy separation method proposed by Ferrer Balas et al. [27] was applied in order to analyze exclusively the initiation process and to overcome the unstable crack propagations problem presented in the studied samples. The processing conditions were selected in order to promote an “in situ” microfibrillation of rPET-O without the need of a “precursor” morphology in the raw material, allowing to explore this potential ability as a function of rPET-O content. The induced morphology of blends was also studied using scanning electron microscopy and dynamic rheological experiments.

## 2. Materials and Methods

### 2.1. Raw Materials

A commercially available post-consumer polypropylene (rPP), predominantly reclaimed from packaging waste, was obtained from Quality Circular Polymers (Limburg, The Netherlands) under the commercial name QPC^TM^ EXPP 152A (MFI (230 °C, 2.16 Kg) = 17.5 ± 0.3 g·10 min^−1^). As minority phase in the blends, a PET recovered from opaque bottles (rPET-O) was supplied by Suez RV Plastiques Atlantique (Bayonne, France) under the trade name Floreal. According to a previous study, rPET-O has a TiO_2_ content of 1.45 ± 0.05 wt.% with a particle size distribution between 152 and 370 nm and a weighted average size of 261 nm [20].

### 2.2. Blends’ Preparation

Raw materials were processed by the Institute of Analytical Sciences and Physico-Chemistry for Environment and Materials of the University of Pau and the Adour Region (IPREM-UPPA), which is a partner of the RevalPET project, according to the following two extrusion steps: homogenization process of the abovementioned physically mixed flakes in order to obtain regular pellets (referred to as rPET-O) and melt blending process of three different rPP/rPET-O blends containing 10, 20, and 30 wt.% of rPET-O (denoted as 90/10, 80/20, and 70/30, respectively, hereinafter).

Prior to both processing steeps, the materials were dried at 80 °C for 24 h and processed by a co-rotating twin screw extruder (LABTECH company Ltd., Samut Prakan, Thailand) with a screw diameter of 16 mm (L/D = 40) and ten heating zones. In both cases, the temperature profile from the feeding zone to the die was in the range of 150–265 °C, respectively, and the screw speed was set to 150 rpm. The take up conditions were set in such way so as not to promote any intentional precursor fibrillated morphology. In order to avoid thermal degradation during melt blending, 0.5%wt of an hydrolytically stable organophosphite processing stabilizer (Irgafos 168) was added to the physical solid mixing step of raw materials.

### 2.3. Processing and Recycled Blends Sheets Manufacturing

From pelletized blends, 30 m of sheets (nominal width: 100 mm; nominal thickness: 0.6 mm) of each rPP/rPET-O formulation as well as raw materials (rPET-O and rPP) were manufactured by extrusion-calendering. Prior to processing, pellets were dried for 4 h at 120 °C in a PIOVAN hopper-dryer (DSN506HE, Venice, Italy) with a dew point of −40 °C and kept under the same dry temperature over the whole processing time. A co-rotating twin-screw extruder Collin Kneter 25 × 24D (COLLIN GmbH, Ebersberg, Germany) with a screw diameter of 25 mm (L/D = 36) and seven heating zones was operated using the temperature profile indicated in Table 1 with screw speed fixed at 55 rpm. During the whole extrusion-calendering process, an N_2_ blanket was introduced into the feeding zone and vacuum was applied in the metering zone. The chill roll temperature of the calendering system (Techline CR72T, COLLIN, Ebersberg, Germany) was set to 50 °C and the take-up speed was set to 0.5 m.min^−1^ in order to minimize molecular orientation.

### 2.4. Morphological Characterization

In order to reveal the distribution of the *rPET-O* phase in the extruded sheets, a morphological characterization was carried out on cryogenically fractured surfaces, on both machine direction (MD) as well as transversal direction (TD) using a JOEL-JSM-7001F scanning electron microscope (JOEL Ltd., Tokyo, Japan) operated at 2 kV. Prior to observations, all samples were sputter coated with a thin platinum–palladium (80:20) layer.

### 2.5. Rheological Characterization

Dynamic rheological measurements were performed using an AR-G2 rheometer (TA Instruments, New Castle, DE, USA) in a parallel plate (25 mm) configuration under N_2_ atmosphere with a constant gap of 0.63 mm at 190 °C. This temperature is above the melting point of rPP and yet low enough to prevent the rPET-O disperse phase from melting. Small amplitude oscillatory experiments were carried out in the angular frequency range 1 < *ω* < 623 rad·s^−1^ at 2% strain (linear viscoelastic regime). To ensure sufficient data in the terminal regime of all the samples, creep-recovery experiments were performed using fresh samples. The compliance data, J(t), were converted to dynamic measurements using the NLREG method [28,29]. Prior to testing, samples were vacuum-dried at 120 °C overnight.

### 2.6. Differential Scanning Calorimetry (DSC)

It is well known that the final mechanical and fracture properties of polymeric composites are significantly dependent on the components’ crystallinity developed during processing [30]. Thus, the “as received” or thermal behaviour of the as extruded-calendered sheets was assessed thought DSC experiments using a MDSC Q2000 instrument (TA Instruments, New Castle, DE, USA) under a dry N_2_ atmosphere. Standard aluminum pans were used to seal 6–7 mg of each sample, which were then subjected to a heating scan from 30 to 270 °C at a heating rate of 10 and 2 °C.min^−1^. From the enthalpy of each thermal process related to the semicrystalline phases, the degree of crystallinity, *X_c_*, was calculated as follows:(1)$$X_{c}\left(\%\right)=\frac{\Delta H_{m}-\Delta H_{cc}}{\Delta H_{m}^{0}\times \phi }\times 100$$
where Δ*H_m_* is the melting enthalpy, Δ*H_cc_* is the cold crystallization enthalpy, *φ* is the weight fraction of the considered polymer phase, and $\Delta H_{m}^{0}$ is the theoretical melting enthalpy for a 100% crystalline PP (207 J·g^−1^ [31]) or PET (140 J·g^−1^ [16,32]).

### 2.7. Tensile Testing

The uniaxial tensile behaviour was assessed according to ISO 527 using a GALDABINI SUN 2500 testing machine (GALDABINI, Cardano al Campo, Italy) equipped with a 1 kN load cell. Type 1BA dumbbell specimens according to ISO 527-3 [33] were extracted from the center of the calendered blend sheets parallel to MD. The specimens were tested at a constant crosshead speed of 1 mm.min^−1^ at room temperature (RT). The Young’s modulus, E, was determined employing a video extensometer (OS-65D CCD, Minstron, Taipei, Taiwan) and using an initial length between marks in the slender section L_1_ = 25 mm. E; yield stress, σ_y_; yield strain, ε_y_; and strain at break, ε_b_, were determined from the engineering stress–strain curves. In each case, the mean value (and its standard deviation) of at least seven valid measurements was reported.

### 2.8. Fracture Behaviour

The fracture behaviour of blend sheets was evaluated employing the same universal testing machine described previously using deeply double-edge notched tensile (DDENT) specimens extracted from the center of the sheets, parallel to MD, as depicted in Figure 1a, with the following dimensions: overall length L = 100 mm, distance between grips L_G_ = 60 mm, width *W* = 50 mm, and nominal thickness *t* = 0.6 mm. Nine ligament lengths, *l*, varied in a broad range of 10–22 mm with a step of 1.5 mm were tested and repeated at least four times. In each test specimen, two equidistant video extensometer marks, L_V_, were set equal to the corresponding nominal ligament length for each sample. Before testing, notch pushing technique with a fresh steel razor blade in DDENT specimens was applied [34]. Tests were performed at RT at a constant crosshead speed of 10 mm.min^−1^.

After testing, the height of the plastic deformation zone, h, surrounding the fracture zone (Figure 1b) and the real ligament length (after notch sharpening) were measured by a binocular lens microscope (Carton, Pathumthani, Rangsit, Thailand). For selected ligament length of each material, EWF test was performed, coupling a video monitoring system (Schneider Kreuznach, Bad Kreuznach, Germany) merging to an optical strain measuring system (ARAMIS, GOM GmbH, Braunschweig, Germany). The advanced strain analysis was accomplished using digital image correlation (DIC) in both notch tips within an acquisition interval of 10 images per second.

The total work of fracture, *W_f_*, was calculated from the integral of the recorded load-displacement (*L–d*) curve for each tested ligament length. Data analysis was performed following the methodology suggested by the standard protocol of the Technical committee 4 of the European Structural Integrity Society (ESIS-TC4) [35]. In order to gain more insight into the crack initiation process, the same obtained data were analyzed considering the partition energy methodology suggested by Ferrer-Balas et al. [27]. Briefly, the total work of fracture, *W_f_*, is separated in two components: *W_I_* (irreversible initiation process involving yielding, crack-tip blunting, and necking) and *W_II_* (crack propagation and extended necking in the out-of-crack-plane zone). By dividing each component by the fracture surface (*l**t*), the specific terms were obtained. Thus, the equations can be written as follows:(2)$$W_{f}=W_{I}+W_{II}$$
(3)$$\frac{W_{f}}{lt}=w_{f}=w_{f,I}+w_{\;f,II}$$
where *w**_f,I_* and *w**_f,II_*are the initiation and propagation specific total work of fracture, respectively. Each term adopts the former EWF data reduction scheme and graphical analysis, which, for the initiation stage, could be written as follows:(4)$$w_{f,I}=w_{e,I}+\beta w_{p,I}l$$
where *w**_e,I_* and *βw**_p,I_* are the specific essential (per ligament area) and non-essential (per volume unit) work of fracture in the initiation term, respectively. From both, the one with clear significance is *w**_e,I_* as it represents the specific work done at the initiation of the crack propagation that involves yielding, crack tip blunting, and necking of its propagation plane.

In this partitioning proposal, the absorbed elastic energy in the necked specimen is supposed to be released during the crack propagation and extended necking process, and it is not included in the initiation work. Thus, the splitting of total work is made on the basis of the elastic stiffness (slope of the initial part of the load–displacement curve). The key point in this methodology is to determine the onset crack propagation during the test and locate this in the load–displacement curve registered. This onset determination with the help of a DIC analysis and how the partition is made was first proposed by Ferrer-Balas et al. [27] and an example is presented in Figure 2. In all the blends, this onset coincides with the inflection point after yielding is reached, so the actual point was determined by the first derivative analysis of L_v_, marking the displacement–time *(d–t)* curve.

## 3. Results

### 3.1. Morphologies of rPP/rPET-O Blends

Figure 3 shows SEM micrographs of the different cryofractured samples in both flow directions and in regions close to the outer layer and the center of the analyzed sections. As can be seen, independently of the formulation considered, the expected two-phase structure was observed, confirming the basic immiscible nature of the rPP/rPET-O blends.

Moreover, as can be seen on the cryofractured surfaces, voiding and particle debonding occurred during fracture, thus indicating a poor interfacial adhesion between both phases. For blends in the composition range 10–20 wt.% of rPET-O, a typical sea-island morphology is revealed, where submicronic and near-spherical rPET-O droplets are relatively homogeneously dispersed in the rPP matrix. As reported in Figure 3, rPET-O droplets exhibited a larger weight-average diameter as the content increased to 20 wt.%.

A detailed inspection in the MD direction of the 80/20 blend reveals that a morphological gradient along the thickness of the sample defines a “skin-core” overall morphology. The “skin” (outer faces of the sheet) is characterized by pseudofibrils with irregular cross-section of the rPET-O phase, with the axis oriented in the direction of the flow. At about 150 μm from the outer layer, a much more heterogeneous morphology, losing the pseudofibrillar character, generating an oblong globular morphology oriented in the direction of extrusion flow. When the rPET-O concentration increased to 30 wt.%, a fine fibrillar morphology mainly oriented in the MD direction was observed. The average diameter of the rPET-O fibrils in the resulting in situ microfibrillar composite (MFCs) was as low as 1.7 μm and all of them demonstrated high aspect ratios.

The morphological situation obtained as a function of rPET-O content can be understood considering that the development of the final morphology in polymer blends is highly dependent on the deformation events that occur in the minority phase, especially in the final sections of the extrusion process (die). During the calendering process, the material is subjected to an elongational and shear flow fields, producing a morphological gradient parabolic-type along the thickness of the calendered sheet, where the maximum shear occurs near to the “skin” zone and the minimum in the center one. On the other hand, the development of microfibrillar structures in immiscible polymer blends is promoted by the coalescence effect during drawing. When the microfibers featured a high aspect ratio, they could break during processing if a high shear rate develops [36]. In reference to the studied blends, the minority proportion of the secondary phase increases the probability to break the induced microfibers in the “die-land”. However, by increasing this proportion, the elongation capacity of the in the microfiber increases without reaching breakage. The generation of this microfibrillar structure and its evolution can be attributed to the high shear field occurring in the “die-land” area, after the pool or “manifold” of the head, which, together with an elongation field by convergent flow, largely conditions the viscosity relationship between the phases, favoring the rheological conditions for the appearance of rPET-O fibrillar structures in those regions with local greater shear between phases.

On the contrary, for the lowest rPET-O content (90/10 blend), the morphology generated is the droplet or of the double emulsion type. In this case, the stress field generated combined with the low content of dispersed phase can generate filaments of dispersed phase with high stability, reaching their rupture by the interfacial instability mechanism, either of the Rayleigh type or of the end pinching type. Depending on the relaxation time involved in the fluid after the applied stress field ceases, these can coalesce and generate a larger globular morphology [37].

### 3.2. Rheological Properties of the rPP/rPET-O Calendered Sheets

It is noteworthy that the analysis of rPP/rPET-O micrographs suffered from the intrinsic two-dimensional feature of the image coupled with the poor separation between both polymers. Thus, the possibility of misinterpreting the blend morphology is expected. To substantiate this experimental issue, a rheological characterization was carried out to gain an insight into the microstructures of the formulations under consideration. In fact, it is well known that the relaxation dynamics of the macromolecular chains in composite melts are significatively dependent on the secondary phase deformability, its shape and particle size, the level of dispersion, as well as the interfacial strength between polymers [38]. Therefore, to verify the morphological situations observed by SEM, no experiments were performed in pure rPET-O sample because, to know its reinforcing effect, it must be in the rubber-like state when the matrix is melted.

Figure 4a,b present the angular frequency dependence of the complex viscosity, |η*|, and the storage modulus, G′, respectively, above the melting temperature (*T_m_*) of rPP, but below the *T_m_* of rPET on the rPP/rPET-O blends.

Figure 4c depicts the semilogarithmic plot of the phase angle, *δ*, as a function of the complex modulus, G*, also known as the Van Gurp–Palmen plot. Recall that a purely viscous behaviour is characterized by *δ* = 90°, while a purely elastic melted material exhibits a *δ* value of 0°. rPP exhibited the typical pseudoplastic fluid behaviour, i.e., the continuously increasing *δ* up to a limiting plateau value at 90° as G* decreased was indicative of a predominant liquid-like behaviour (G′′ > G′). As can be seen in Figure 4, 90/10 blends exhibited a similar rheological behaviour as compared with rPP alone. This behaviour indicated that the presence of 10 wt.% of unmelted submicronic rPET-O droplets is not sufficient to physically constrain rPP chains to flow and confirm the lack of intermolecular interactions between both polymers.

Drastic changes in the rheological behaviour were observed when 20 and 30 wt.% of rPET-O were added to rPP. In fact, |η*| as well as G′ values gradually increased with the rPET-O content over the whole experimental range. Considering the Van Gurp–Palmen plots and moving from the terminal regime, a common feature that could be observed was that both blends developed an inflection point to a minimum, then increased again up to a maximum, and finally decreased as the dynamics reached the glassy regime. This behaviour has been widely reported in copolymers and binary blends [39,40]. Regarding the 80/20 formulation, it can be observed that a shoulder is observed in the |η*| as well as in the G′ curves before reaching the terminal region, which is shifted to lower ω. In the Van Gurp–Palmen plots, it was observed that the minimum occurred when G* was around 6 × 10^2^ Pa, which is close to the plateau modulus (Figure 4b).

This behaviour can be attributed to the restriction of the rPP phase mobility in the vicinity of the solid rPET-O domains. That is, while the unperturbed rPP chains are free to relax by relatively fast self-diffusion processes, the lowered thickness of the interlayer between rPET-O domains restricted the molecular mobility of the rPP chains surrounding particles that must relax through other longer mechanisms. In the low ω region, the retarded macromolecular relaxation enhanced the solid-like behaviour of the melt, as confirmed by the higher G′ value than rPP. These behaviours are often reported in blends with a droplet morphology [41,42].

The 70/30 blend exhibited the characteristic rheological behaviour of structured fluids. As ω decreased, the |η*| increase is indicative of an apparent yield stress. The increasing time required to flow weakened the low-ω power law dependence and G′ values tended to reach a plateau value. All these changes indicated a transition from liquid to solid-like behaviour. When the Van Gurp–Palmen plot is considered, between the terminal regime and the rubbery plateau minimum, G* was around 9 × 10^2^ Pa, with *δ* values lower than 45°, implying that the elastic component of the melt exceeds the viscous one. The results can be attributed to the phase separation between elements having a different modulus as well as the mechanical entanglement produced by the rPET-O microfibrillar structure, which acts as a reinforcement agent, reducing the phase angle before reaching the terminal regime.

Such behaviour is often related to composites exhibiting a physically entangled fibrillar network and percolation threshold effect of the second phase, which, in the present study, would have been created by the topological interactions of the long rPET-O microfibrils [40,43]. Based on the conclusions previously reported, the gradual curve step phenomenon observed in the Van Gurp–Palmen plot as the rPET-O content increases must be attributed to the increasing solid-like behaviour of the melt owing to significant changes in the relaxation mechanisms as a function of the developed morphology.

### 3.3. Thermal Properties

Figure 5 presents the DSC heating scans at 10 and 2 °C.min^−1^. The main transition temperatures related to the crystalline phase of both rPP and rPET-O: the melting temperature (*T_m_*) and melting enthalpy (Δ*H_m_*) of the whole polyolefin phase (rPP), the cold crystallization temperature (*T_cc_*), the melting temperature (*T_m_*), and the degree of crystallinity (*X_c_*) of rPET-O phase are reported in Table 2.

Figure 5a shows that the rPET-O sample displayed the typical thermal transitions of predominantly amorphous PET and was confirmed by the calculated *X_c_*. In contrast, under similar processing conditions, the fastest crystallization kinetics exhibited by rPP allowed to manufacture semi-crystalline rPP sheets exhibiting a well-defined melting transition around 163 °C. It is important to note that a small endotherm (indicated by an arrow in Figure 5a) was detected at 126 °C prior to the main endotherm of rPP. This melting behaviour is in line with the technical data sheet of the selected rPP grade and ascribed to the melting of semi-crystalline polyethylene (PE) impurities. Thus, no attempts for *X_c_* of rPP phase were made and, only for comparison purposes, Δ*H_m_* was determined.

Independent of the morphology developed, the thermal behaviours of all the rPP/rPET-O blends are similar and corroborate the inherent immiscibility of the considered blends. Both phases melt separately and no significant changes in their *T_m_* are observed. It is noteworthy that, for the 10 °C.min^−1^ heating scan, the cold-crystallization transition of rPET-O overlaps with the melting of the PE impurities, making the calculation of *X_c_* developed by rPET-O phase impossible during processing. On this basis, fresh DSC experiments were performed at 2 °C.min^−1^ in order to separate the cold crystallization process of rPET-O to the melting transition of PE impurities. As can be seen in Figure 5b (*T_cc_* indicated by an arrow), satisfactory results were obtained and it was calculated that rPET-O developed a *X_c_* around 29% in all blends. Additionally, it was observed that the whole rPP phase featured similar Δ*H_m_* values independently of the formulation considered. This fact could be an indication that this phase developed the same crystalline amount in all blends.

In relation to the rPET-O phase in blends, the increase of crystallinity degree as compared with pure rPET-O suggests that, at the processing temperature, it has been subjected to a hot stretching process in the rubbery state when the melted blend crosses the convergence zone of the die, thus promoting the thermo-mechanical effect of strain-induced crystallization.

### 3.4. Mechanical Behaviour of the Extruded Sheets: Tensile Test

Representative engineering stress–strain (σ–ε) curves of rPP and blends including the appearance of the breaking zone are shown in Figure 6a,b, respectively.

All of them presented a local maximum associated with a yield point. Nevertheless, the post-yielding behaviour differs significantly. While the addition of 10–20 wt.% of rPET-O preserved the characteristic inception and stable neck propagation of a ductile polymer, being smaller for 80/20 blend, it was totally suppressed in the presence of 30 wt.% of rPET-O, thus generating a ductile σ–ε curve without a cold drawing stage (Figure 6b). During testing, visual observations suggested that the rPET-O fibrillar structure promoted the formation of multiple crazes, hence preventing the propagation of a stable neck and giving rise to a significant decrease in the elongation at break, ε_b_.

In this case, the reduction in the density of effective entanglements, as a consequence of the microfibrillar morphology generated when 30 wt.% of rPET-O was added, decreased the density of the entanglement network in the rPP. As a result, the necessary conditions in the local stress field for micro-void nucleation and craze generation are promoted [44].

Such mechanical behaviour was confirmed by SEM observations on the tested samples, as shown in Figure 7. While the rPP fracture surface in the 80/20 blend showed large plastic deformation, the fracture surface of rPP in the 70/30 blend was smooth, typical of a brittle material. In the latter case, the microfibril surface was very smooth and no rPP particles shifted to the rPET-O component, confirming the lack of adhesion between components; however, the fibrillar structure of the system was preserved.

Table 3 shows the tensile properties obtained from these tests. In order to compare the blends under equal crystalline conditions of rPET-O phase, a set of rPET-O tensile samples were subjected to a recrystallization process (30 min. at 120 °C). After this thermal treatment, the samples developed a *X_c_* = 30%, similar to those observed in the blends (c.f. Table 2).

In the elastic range, it was found that E gradually increased, with the rPET-O content being 36% higher than rPP for the 70/30 blend (Table 3). Figure 8a shows the tendency of E considering the additive rule of mixtures, ARM, exhibiting an evident positive deviation for the 70/30 blend. This positive result could be explained by the fact that ARM predictions are based without considering the geometry and orientation effects of the dispersed phase [45]. The results suggested that the rPET-O microfibrils developed during calendering are long enough to act as reinforcing elements in the elastic range of the blends owing to their higher load-bearing capacity. During loading, the matrix exerts pressure on the fibrils, producing a considerable levels of frictional forces, which led to an increase in E [12].

Regarding the yield point and considering the standard deviations in the σ_y_ values, it remained practically unaffected in the blends. That is, this parameter did not exhibit any dependence by the addition of rPET-O in the polymeric systems (Table 3), confirming that, in the strength field, this response is governed by the rPP matrix. However, the deformation associated with this point revelated an evident dependence of ε_y_ on the rPET-O content in blends, thus defining a highlighted negative deviation from the ARM (Figure 8b).

The results suggest that the de-cohesion and cavitation of the dispersed phase at the yield point seem to be the micromechanism to release the triaxial stress state prior to the following stress drop.

### 3.5. Fracture Behaviour: Essential Work of Fracture Analysis

The representative load (F) versus relative displacement (video extensometer measurement, d) curves for three different ligament lengths of the analysed rPP and blends are shown in Figure 9. Independent of the ligament length, it was observed that, for a given material, the curves accomplish self-similarity evolution, which would initially validate the application of the EWF technique. In the case of rPP, it developed a characteristic curve of an excessive blunting of the notch tip prior to its propagation, an aspect that limits the validity of the EWF application. As can be seen in Figure 9, when the proportion of rPET-O increased, the tearing stage (point indicated as onset of crack propagation using the video monitoring system) became more irregular, being completely suppressed for the 70/30 samples.

It is important to note the highest value of the maximum load registered by this last blend. At uniaxial loading, this could highlight the effect of dispersed phase fibrillation as a local reinforcing agent. The appearance of the process zone (ligament region) after testing of the studied materials is depicted in Figure 10. In all cases, a whitening of the area adjacent to the ideal plane of crack propagation (the plane that defines the collinear pre-notches) is observed. This zone gradually blurs and takes on a straited appearance as the rPET-O content increases.

This region was considered to determine the geometric parameter *β*, related to the size of the process zone. When 20 and 30 wt.% of rPET-O was added, the propagation after plastic, the collapse of the ligament became discontinuous and irregular. This can be attributed to the high heterogeneity in the morphology obtained (fibrils and droplets). A preliminary determination of the thickness of the process zone revelated that no level of striction developed. Thus, this whitening could be attributed to an extensive and irregular cavitation of the rPET-O phase with minimal contribution of plastic flow.

As described in Section 2.8, by integrating the F–d curves from all tested ligaments (c.f. Figure 9), it was possible to represent *w**_f_* vs. *l*, easily obtaining the *w**_e_* and *βw_p_* parameters from the best linear regression of the set values, as can be seen in Figure 11. Following the ESIS protocol [35], h of the necked zone was determined for each tested sample. After testing, it was observed that the plastic deformation zone developed a parabolic-like form (c.f. Figure 1 and Figure 10). Then, from the slope of the fitted line of the set of values represented in a graph h vs. *l*, the shape factor related to the plastic deformation zone, β, was estimated as follows [46]:h = 1.5 × *β* × *l*(5)

Table 4 summaries the numerical parameters obtained after applying the EWF analysis. In all cases, the correlation coefficient (R^2^) exceeds 0.92, confirming the quality of the data. It is important to mention that both specific fracture parameters determined for rPP are in line with those reported in the literature for an ethylene propylene block copolymers (EPBC) with 7.4 wt.% of ethylene content [47], as could be expected owing to observations obtained by DSC (c.f. Figure 5).

It can be seen that both essential (*w**_e_*) and non-essential (*βw_p_*) terms of rPP decreased between 50 and 80% and 20 and 75%, respectively, when rPET-O was added. This highlights the low reinforcement capacity of the fibrillar morphology that could have been generated in the sheets with higher rPET-O contents. However, it must be considered that the EWF analysis methodology considers the entire process: the onset of crack propagation and final tearing. Therefore, *w**_e_* is highly influenced by the final stage. Considering the low adhesion between components, the phase decohesion process started at the tip of the sharp notch before the onset of crack propagation. Then, during crack propagation in the collinear plane between notches, it encountered cavitated areas, facilitating the process. This aspect is much more evident for those morphologies with fibrillation.

Considering the *β* parameter, related to the size area where the energy dissipation processes occur through deformation (external to the propagation plane), this showed a clear increase in blends with respect to rPP, reaching a maximum (more than double) when adding 30 wt.% of rPET-O. The results suggest that the “area of influence” promoted by the fibrillar structure during deformation and cavitation is greater. Therefore, the method does not allow an adequate analysis of tear behaviour owing to the morphology and degree of adherence between phases. For this reason, to analyze exclusively the fracture behaviour up to the onset of crack propagation, the initiation contributing term related to the partition energy method (c.f. Figure 2) was used.

Figure 12 displays the initiation-related specific total work of fracture, *w**_f,I_*, versus ligament lengths for all blends. Table 5 summarize the initiation-related specific EWF parameter, *w**_e,I_*, of rPP/rPET-O calendered sheets.

Significant differences in the initiation parameter were observed. The addition of a small amount of rPET-O significantly decreased the resistance to stable crack initiation, *w**_e,I_*, of rPP (used strictly for comparison purposes and not shown for the sake of clarity: 56 ± 0.8 $\mathrm{kJ}\cdot m^{-2}$). It may be attributed to the spherical shape of rPET-O, which acted as microstructural defects, promoting matrix cavitation processes ahead of the crack and crack propagation at low energy levels. Similar behaviour has been widely reported in the literature for polypropylene-based binary polymer composites [25,48].

However, the 70/30 sample featured a considerable increase (more than double) as compared with the other blends. In fact, when comparing the initiation-related and global essential parameters, it was observed that, from 16 kJ·m^−2^, about 41% was used during the initiation stage, spending much more energy than other blends (Table 5). This increment could be ascribed to the microfibrillar structure developed during calendering owing to the rPP-based processing temperature of the sheet (215 °C). These fibers apparently reached a length long enough to act as a reinforcing element. Then, when starting the fracture test, frictional forces along the fiber length and the matrix as well as the physical entanglements improved the resistance to crack initiation, thus increasing the *w**_e,I_* value.

## 4. Conclusions

Under the processing conditions used in this study to obtain calendered sheets, the obtained results lead to the following conclusions. According to the morphological observations, the presence of a small amount of rPET-O developed a complex morphological situation: dispersed globular shape in the 90/10 blend and morphological gradient defined by skin (fibrillar)–core (globular) in the 80/20 blend. Relating morphology with rheological studies by applying the Van Gurp–Palmen model, the potential scope of the in situ microfibrillation technique appears to be greatly enhanced when 30 wt.% of rPET-O was added. However, it is essential to achieve an improvement in adhesion between phases.

From uniaxial tensile tests, it was observed that the yield stress was not affected by the addition of rPET-O. Interestingly, the microfibrillar structure increased the stiffness of the 70/30 blends by up to 36% as compared with pure rPP. According to the additive rule of mixtures, the Young’s modulus showed a linear increase in the longitudinal direction of load, indicating that rPET-O is highly susceptible to enhance the rPP stiffness with a spherical or an oblong globular morphology. However, the in situ fibril structure promoted an improvement in E, increasing up to 7%.

In reference to the EWF technique, the morphological situation generated in blends and adhesion level between phases does not allow an adequate analysis of the tear behaviour owing to the high instability produced during the tearing stage. When analyzing the specific EWF related to initiation, *w**_e,I_*, it was found that the microfibrillar structure developed during calendering on 70/30 blends had a reinforcement effect and enhanced the resistance to crack propagation, increasing the load bearing capacity of a pre-cracked body owing to mechanical interactions between fibers and matrix.

Considering that both rPP and rPET-O raw materials come from wastes and contain intrinsic additional elements (PE as impurities and TiO_2_ as filler, respectively), the MFC structure generated in the 70/30 blend after the blending process gave rise to an increase in the Young´s modulus and appreciable improvement in the initiation-related specific work of fracture even without the presence of compatibilizing agents. Based on the objectives of the circular economy strategy adopted for the European Commission, these results contribute to the “zero-waste” objectives by increasing the recycle rate of PET-O. Therefore, a solution to a potential future problem is proposed, as this material is being discarded from current recycling processes.

## Figures and Tables

**Figure 1 polymers-13-02360-f001:**
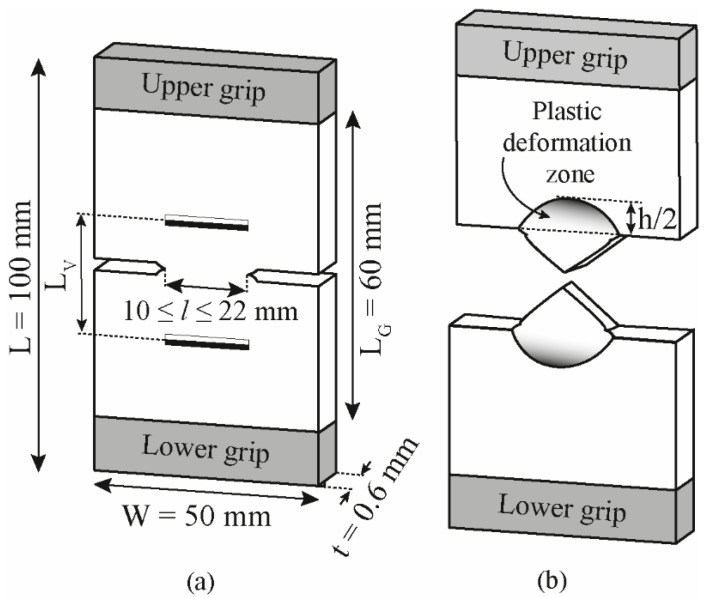
Schematic representation of the deeply double-edge notched tensile (DDENT) geometry with sample dimensions (**a**) before and (**b**) after testing.

**Figure 2 polymers-13-02360-f002:**
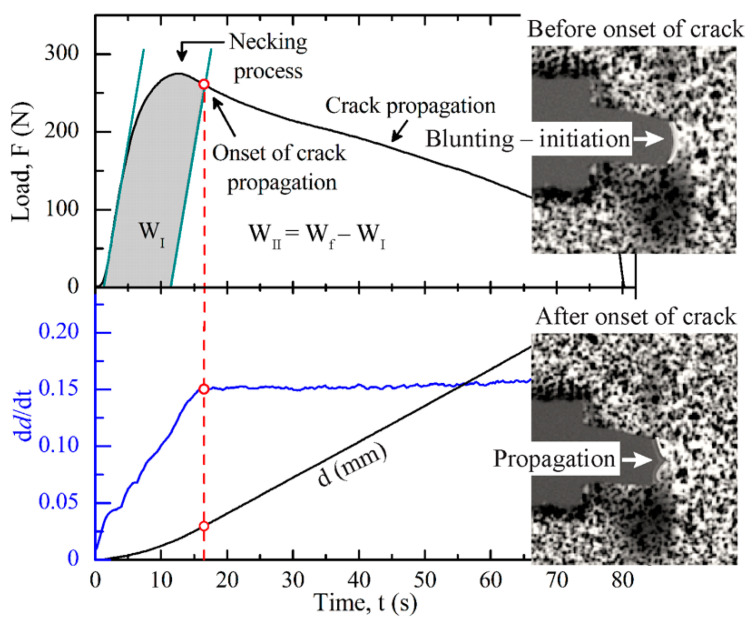
Energy partition illustrated in a load–time curve of 90/10 blends obtained from EWF tests. Also shown below is the derivative analysis of the strain rate of the tested sample. Insets: the notch tip zone images taken an instant before and after the onset of crack propagation.

**Figure 3 polymers-13-02360-f003:**
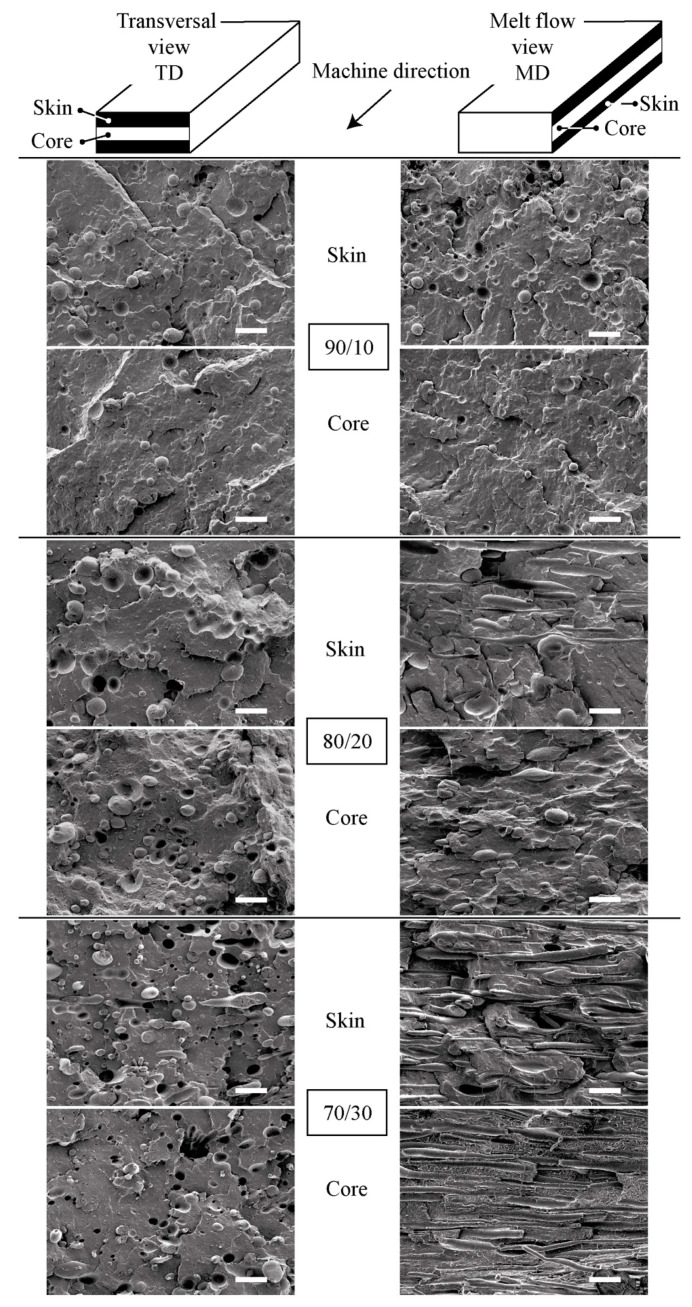
SEM micrographs (×1500) of the cryofractured surfaces of rPP/rPET-O blends in the directions and positions indicated. Scale bars = 10 μm.

**Figure 4 polymers-13-02360-f004:**
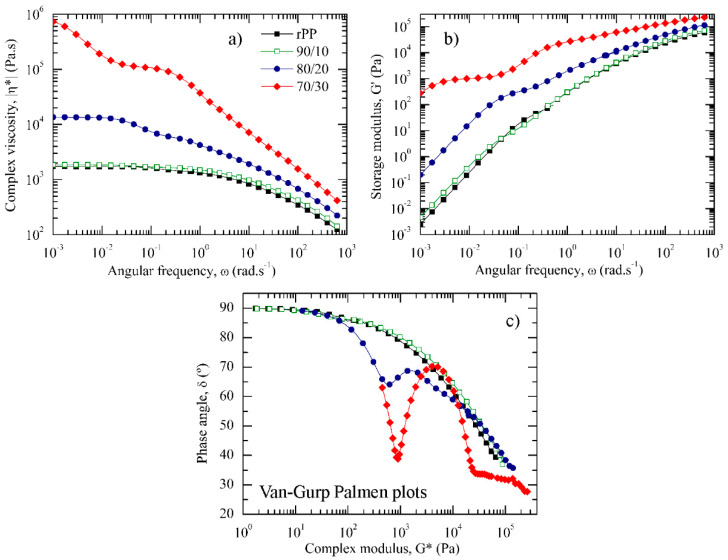
*ω* dependence at 190 °C of the (**a**) $\left|\eta^{*}\right|$ and (**b**) G′. (**c**) Van Gurp–Palmen plots at 190 °C.

**Figure 5 polymers-13-02360-f005:**
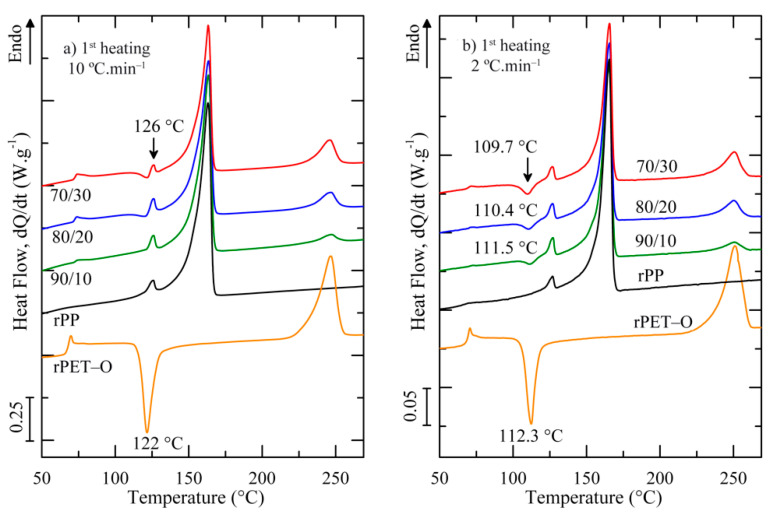
First heating DSC scans of rPP/rPET-O blends at (**a**) 10 °C.min^−1^ and (**b**) 2 °C.min^−1^.

**Figure 6 polymers-13-02360-f006:**
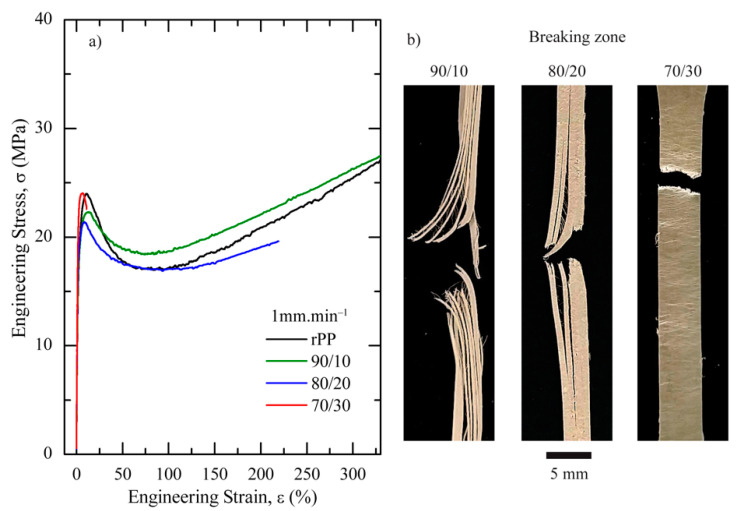
Effect of the rPET-O content on the mechanical properties of rPP/rPET-O sheets: (**a**) representative engineering stress–strain curves and (**b**) visual aspect of the fracture zone.

**Figure 7 polymers-13-02360-f007:**
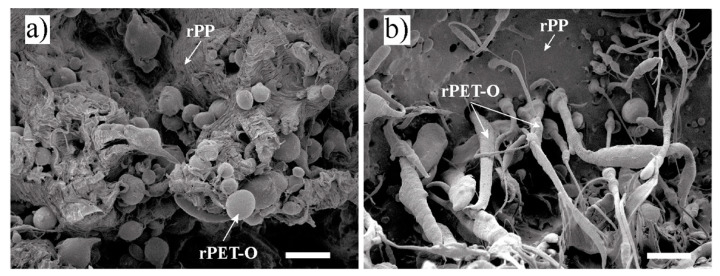
SEM micrographs of the post mortem fracture surfaces (×1500) of (**a**) 80/20 and (**b**) 70/30 rPP/rPET-O blends. Scale bars = 10 mm.

**Figure 8 polymers-13-02360-f008:**
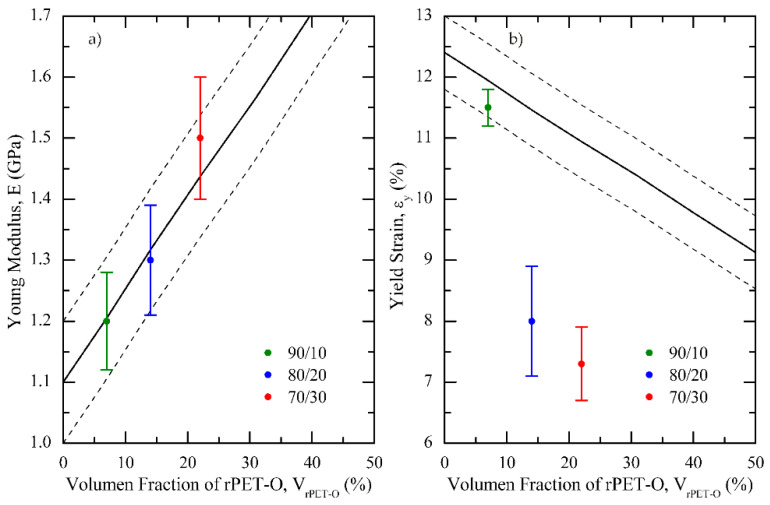
Predicted values by the additive rule of mixtures (ARM) (solid line) and its standard deviation (dashed lines) for (**a**) Young’s modulus (E) and (**b**) yield strain (ε_y_) of rPP/rPET-O samples tested at 1 mm·min^−1^.

**Figure 9 polymers-13-02360-f009:**
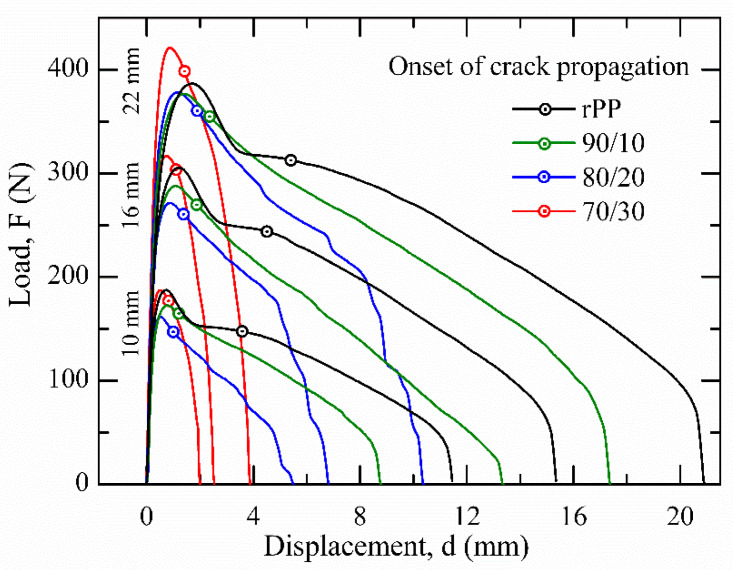
Self-similarity load vs. displacement (F–d) curves of analyzed materials using three different ligament lengths.

**Figure 10 polymers-13-02360-f010:**
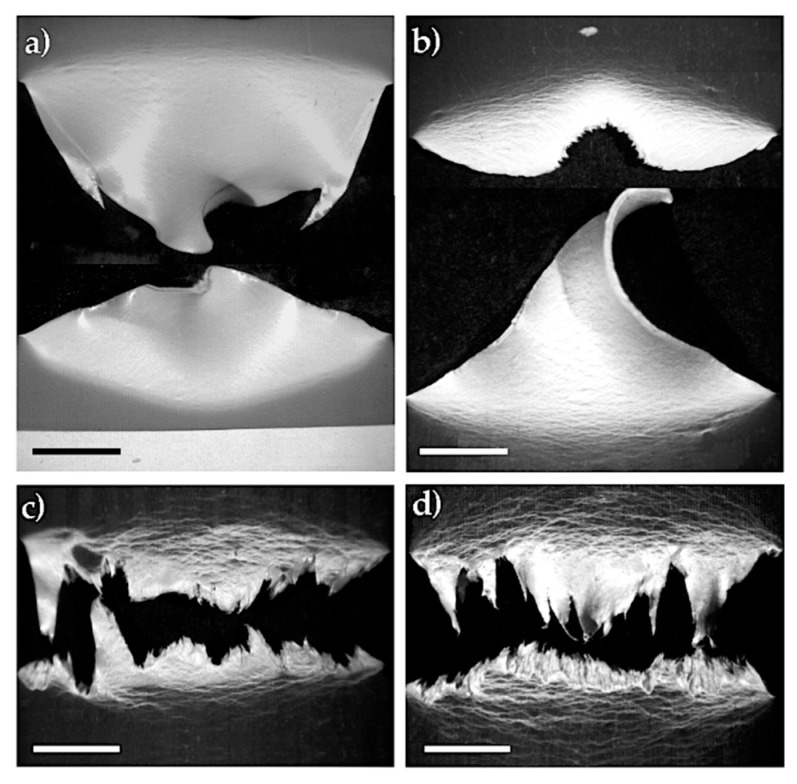
Appearance of the process zones (nominal ligament region: 13 mm) of studied materials after testing: (**a**) rPP, (**b**) 90/10, (**c**) 80/20, and (**d**) 70/30. Scale bars = 3 mm.

**Figure 11 polymers-13-02360-f011:**
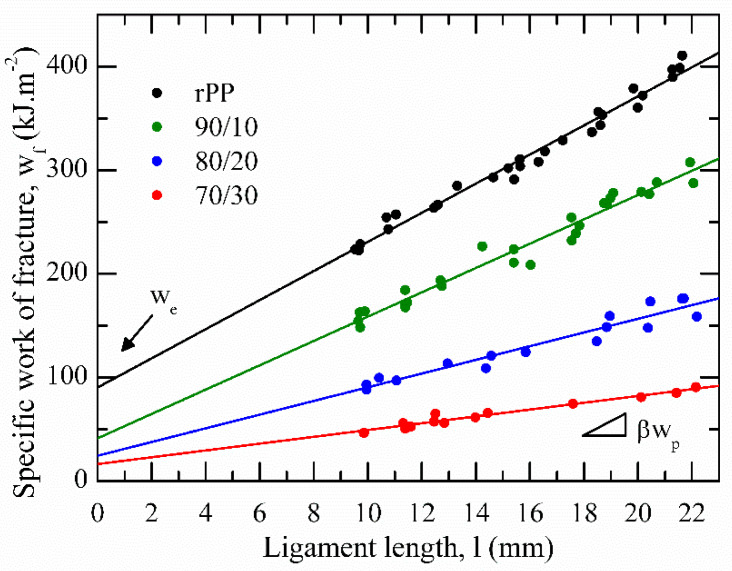
Specific work of fracture as a function of the ligament length for the studied materials.

**Figure 12 polymers-13-02360-f012:**
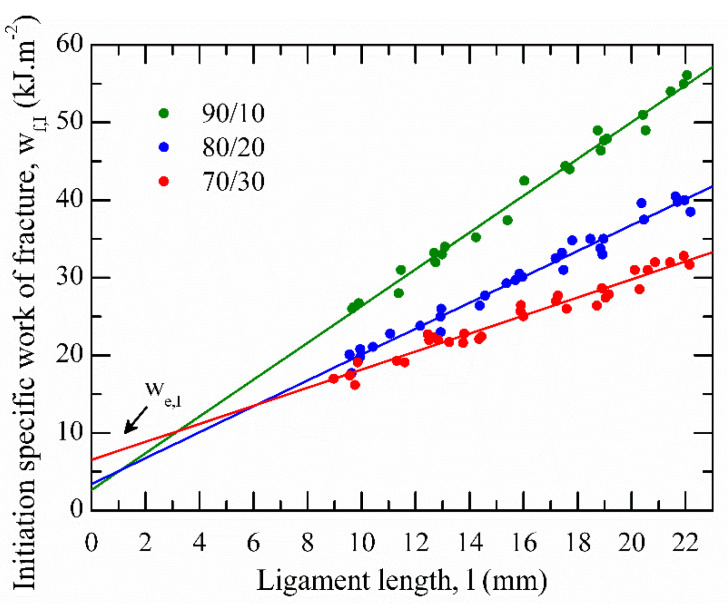
Plot of initiation-related specific work of fracture versus ligament length for the studied polymeric blends. All points are experimental and the fracture parameters were determined by the best linear fit according to Equation (4).

**Table 1 polymers-13-02360-t001:** Designation, composition, and extrusion temperature profile of calendered sheets.

Code	Composition, wt.% (vol.% ^1^)		Temperature Profile (°C) (from Feeding to Die Zone)
	rPP	rPET-O	
rPET-O	0 (0)	100 (100)	180, 215, 235, 240, 240, 245, 245
rPP	100 (100)	0 (0)	140, 180, 195, 205, 215, 215, 210
90/10	90 (93)	10 (7)	140, 180, 195, 205, 215, 215, 210
80/20	80 (86)	20 (14)	140, 180, 195, 205, 215, 215, 210
70/30	70 (78)	30 (22)	140, 180, 195, 205, 215, 215, 210

^1^ Volume fraction of rPET-O (V_rPET-O_) calculated from the density values of each component: ρ_rPET-O_ = 1.35 g·cm^−3^ and ρ_rPP_ = 0.92 g·cm^−3^. V_rPP_ = 1 − V_rPET-O_.

**Table 2 polymers-13-02360-t002:** Thermal properties of the rPP/rPET-O calendered sheets at two different heating rates.

	10 °C.min^−1^					2 °C.min^−1^				
Materials	rPP_phase_		rPET-O_phase_			rPP_phase_		rPET-O_phase_		
	T_m_	ΔH_m_	T_cc_	T_m_	X_c_	T_m_	ΔH_m_	T_cc_	T_m_	X_c_
	(°C)	(J·g^−1^)	(°C)	(°C)	(%)	(°C)	(J·g^−1^)	(°C)	(°C)	(%)
rPET-O	-	-	122	247	8	-	-	112.3	251	8
rPP	163 (126) ^2^	88	-	-	-	166 (127) ^2^	90	-	-	
90/10	163 (126) ^2^	nd	nd	247	nd	165 (127) ^2^	82 ^1^	111.5	251	29
80/20	163 (126) ^2^	nd	nd	247	nd	166 (127) ^2^	85 ^1^	110.4	250	30
70/30	163 (126) ^2^	nd	nd	246	nd	166 (127) ^2^	86 ^1^	109.7	251	29

^1^ Melting enthalpy of the whole polyolefin phase considering exclusively the effective mass of the matrix. ^2^ Melting temperature of PE impurities. nd*:* Not determined.

**Table 3 polymers-13-02360-t003:** Uniaxial tensile properties of rPP/rPET-O samples tested at 1 mm·min^−1^ crosshead speed.

Sample	E	$\sigma_{y}$	$\varepsilon_{y}$	$\varepsilon_{b}$
	(GPa)	(MPa)	(%)	(%)
rPET-O ^1^	2.6 ± 0.1	67 ± 2	5.9 ± 0.1	8 ± 2
rPP	1.10 ± 0.04	24 ± 1	12.4 ± 0.6	420 ± 17
90/10	1.23 ± 0.08	23 ± 1	11.5 ± 0.3	294 ± 53
80/20	1.32 ± 0.09	21 ± 1	8.0 ± 0.9	206 ± 21
70/30	1.5 ± 0.1	24 ± 1	7.3 ± 0.6	13 ± 2

^1^ Mechanical properties of rPET-O sample subjected to a recrystallization process (30 min at 120 °C).

**Table 4 polymers-13-02360-t004:** Total EWF parameters of rPP/rPET-O calendered sheets tested at 10 mm·min^−1^.

Sample Nomenclature	$w_{e}$ $\left(kJ\cdot m^{-2}\right)$	$\beta w_{p}$ $\left(MJ\cdot m^{-3}\right)$	$\beta \times 10^{2}$
rPP	90 ± 6	14.1 ± 0.4	13.4 ± 0.1
90/10	44 ± 4	11.6 ± 0.6	25 ± 2
80/20	24 ± 3	6.6 ± 0.5	26 ± 2
70/30	16 ± 3	3.3 ± 0.2	33 ± 2

**Table 5 polymers-13-02360-t005:** Specific essential EWF parameter related to initiation along with the percentage ratio of initiation/global essential parameters of rPP/rPET-O calendered sheets.

Sample Nomenclature	$w_{e,\;I}$ $\left(kJ\cdot m^{-2}\right)$	$w_{e,\;I}/w_{e}$ (%)
90/10	2.7 ± 0.9	6
80/20	3.4 ± 0.8	14
70/30	6.5 ± 0.7	41

## Data Availability

The study did not report any data.

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
