# Peer review of "Extruded-Calendered Sheets of Fully Recycled PP/Opaque PET Blends: Mechanical and Fracture Behaviour"

_polymers, 2021, doi:10.3390/polym13142360_

Round 1

Reviewer 1 Report

The communication mainly describes the mechanical and fracture behavior of rPP/rPET-O mixtures by uniaxial tensile test and fracture basic work (EWF) method. Meanwhile, the energy separation method was applied, in order to analyze the initiation process and overcome the unstable crack propagations problem presented in the studied samples.

  1. In Section 3.1, the influence of different concentrations on the morphological characteristics of the blends is expounded. However, the morphological differences caused by different positions and different flow directions are not elaborated. It can be supplemented with a brief explanation as appropriate.
  2. In section 4.1 describes the different concentration under the | * | and G’ as , The difference in behavior at different concentrations may be attributed to the restriction of the rPP phase mobility in the vicinity of the solid rPET-O It is suggested that the curve of rPET-O could be added as a reference curve. Meanwhile, an appropriate addition could be made to the reasons for the emergence of the curve step phenomenon in the Van Gurp-Palmen plot.
  3. In Figure 10, the different components are not well distinguished, it is necessary to annotate the rPET-O component corresponding to the image.
  4. the colors representing 90/10 and 70/30 are similar, resulting in insufficient differentiation and presentation, and it is recommended that a color with a greater difference be used.
  5. The relevant description of Figure 11 and the data obtained from Figure 11 for Table 4 is missing and it is suggested that some details could be added.

Generally speaking, I think the paper can be modified and supplemented appropriately in terms of details.

Reviewer 2 Report

The effects of increasing rPET-O concentrations in three different concentrations on the mechanical and fracture behavior of a post-consumer PP (rPP) 93 grade are discussed in this study. Following the EWF concept, the fracture behavior of rPP/rPET-O blends was evaluated on calendered sheets produced using a semi-industrial extrusion-95 calendering process. The MS sounds interesting and can be considered for publication in Polymers after some minor revisions as:

  1. The English language and style are acceptable; a minor spell check is required looking at spelling mistakes and go on;
  2. It is really necessary for a profound literature update since just one from the authors is really new, so at least 50% must be new (last 5 years) showing the importance of the background to the subject theme.

Reviewer 3 Report

Authors of the manuscript present experimental results of the mechanical and fracture behavior of three polymeric blends prepared from two recycled plastics (PP and opaque PET). The raw materials were 2 commercial degrees of recycled post-consumer waste. Sheets from the material were manufactured by a semi-industrial extrusion-calendering process. The mechanical and fracture behavior of the manufactured sheets was analyzed and discussed. The investigations would be very useful for researchers working in polymer field. The paper could be published after the revision:

*Abstract of the manuscript should be more simple and without abbreviations. It should explained clearly for readers the main advantages of the presented investigations as compared with other recyclable polymers.

*In the introduction it should be more clearly presented what is already described in scientific literature about investigations in field of rPP/rPET polymeric mixtures.

*I would recommend also to test a simple rPET without TiO2 in the investigations in order to demonstrate difference of properties of the polymeric mixtures having an additive of TiO2 and of mixtures without TiO2.

*The authors should explain why they have chosen the concentrations in rPP/rPET-O blends containing 10, 20 and 30 wt.% of rPET-O ?

*The authors should explain or demonstrate experimentally why concentrations in rPP/rPET-O blends containing 40 and 50 wt.% of rPET-O are not useful for the investigations ?

*In conclusions of the manuscript it should be more clearly presented the main advantages and disadvantages of the mixtures as material as compared with pure rPP and of rPE.

Round 2

Reviewer 3 Report

After the revision this manuscript could be accepted for publication.